# Biocatalytic Potential of Native Basidiomycetes from Colombia for Flavour/Aroma Production

**DOI:** 10.3390/molecules25184344

**Published:** 2020-09-22

**Authors:** David A. Jaramillo, María J. Méndez, Gabriela Vargas, Elena E. Stashenko, Aída-M. Vasco-Palacios, Andrés Ceballos, Nelson H. Caicedo

**Affiliations:** 1Department of Biochemical Engineering, Universidad Icesi, Calle 18 No. 122–135 Pance, Cali 760031, Colombia; davidalejandro.jaramillo@hotmail.com (D.A.J.); mariaj_0697@hotmail.com (M.J.M.); gabrielavargas2847@gmail.com (G.V.); aceballos@icesi.edu.co (A.C.); 2Universidad Industrial de Santander. Chromatography and Mass Spectrometry Center, Calle 9 Carrera 27, Bucaramanga 680002, Colombia; elena@tucan.uis.edu.co; 3Grupo de Microbiología Ambiental—BioMicro, Escuela de Microbiología, Universidad de Antioquia, UdeA, Calle 70 No. 52–21, Medellín 050010, Colombia; aida.vasco@udea.edu.co

**Keywords:** aromas, basidiomycetes, bioprospection, α-Pinene, *cis*-verbenol

## Abstract

Aromas and flavours can be produced from fungi by either de novo synthesis or biotransformation processes. Herein, the biocatalytic potential of seven basidiomycete species from Colombia fungal strains isolated as endophytes or basidioma was evaluated. *Ganoderma webenarium*, *Ganoderma chocoense*, and *Ganoderma stipitatum* were the most potent strains capable of decolourizing β,β-carotene as evidence of their potential as biocatalysts for de novo aroma synthesis. Since a species’ biocatalytic potential cannot solely be determined via qualitative screening using β,β-carotene biotransformation processes, we focused on using α-pinene biotransformation with mycelium as a measure of catalytic potential. Here, two strains of *Trametes elegans*—namely, the endophytic (ET-06) and basidioma (EBB-046) strains—were screened. Herein, *T. elegans* is reported for the first time as a novel biocatalyst for the oxidation of α-pinene, with a product yield of 2.9 mg of *cis*-Verbenol per gram of dry weight mycelia used. The EBB-046 strain generated flavour compounds via the biotransformation of a Cape gooseberry medium and de novo synthesis in submerged cultures. Three aroma-producing compounds were identified via GC–MS—namely, methyl-3-methoxy-4H-pyran-4-one, hexahydro-3-(methylpropyl)-pyrrolo[1,2-a]pyrazine-1,4-dione, and hexahydro-3-(methylphenyl)-pyrrolo[1,2-a]pyrazine-1,4-dione.

## 1. Introduction

Recently, the market for flavours and aromatic agents derived from natural sources or via traditional techniques of chemical synthesis has grown exponentially; profit margins in this seven-billion-dollar sector have shown an average annual growth of approximately 4.4% [1]. Companies are increasingly turning to biotechnologically produced flavours and aromas, as these methods are cheaper than conventional production techniques and have a lower impact on the environment [2]. Additionally, traditional methods of chemical synthesis are often plagued with issues such as low selectivity, poor yields, and the prevalence of secondary reactions that affect the purity and composition of the final product. These inefficiencies in the production process ultimately lead to significantly higher separation and purification costs [3]. Given these problems, more researchers are turning to new biocatalysts as the primary channel for synthesis, especially for chiral reactions [4]. While this trend is encouraging, it also raises concerns about what can be defined as a natural flavour. International regulations dictate that a natural flavour must be obtained via microbial or enzymatic processes, provided that the raw material is also natural and derived via physical or biotechnological methods [5]. These regulations have become the driving force behind the development of new biotechnological techniques as novel sources for natural volatile flavours, with more than 100 reported compounds commercially available from advanced bioprocesses [6,7].

Biotransformations can be briefly described as chemical reactions catalysed by microorganisms or enzyme systems. These reactions are usually conducted using growing cultures (de novo synthesis), previously grown cells, immobilised cells, purified enzymes, or multiphase systems [8], thereby providing added-value products through bioprocesses. Some studies have reported on the application of biotechnological methods based on microbial transformation reactions [9,10,11,12] or vegetable materials [2] to produce natural aromas. Other studies have focused on the catalytic transformation of carotenoids and monoterpenes such as α- and β-pinene and β-carotenes to produce precursor compounds [13,14,15]. Recently, fungi belonging to the phyla Ascomycota, Zygomycota, and Basidiomycota were shown to biotransform several apocarotenoid flavour agents and fragrance components [1,2,16]. Studies on various types of microorganisms—namely, prokaryotes and eukaryotes—highlighted the ability of basidiomycete fungi to transform the aforementioned compounds [17], as these fungi either produced enzymes that actively oxidised monoterpenes and carotenes [18,19,20] or possessed metabolic pathways for the synthesis of volatile aromatic compounds [21,22]. Many terpenic aldehydes, alcohols, and esters of commercial value were obtained via the oxidation of the above-mentioned precursors or in submerged cultures [11]. These compounds were pivotal for the production of commercial fragrances, as they were extremely volatile and possessed highly desirable organoleptic properties [23]. Additionally, these molecules exhibited biological activity as antioxidants and anti-inflammatory agents [24]. The selective oxidation of α-pinene and β-carotenes produced compounds of industrial interest, such as verbenone and verbenol (from α-pinene) and apocarotenes such as α- and β-ionone (from β-carotenes). Verbenol and verbenone are essential commercial products used as flavouring compounds in pheromone traps [25] and the cosmetic industry [17]. α- and β-ionone are vital for the fragrance and flavour industries, as they have a warm, woody, and fruity aroma that is reminiscent of raspberries with floral notes [26,27]. Synthesising or extracting pure α-ionone from plant sources is not possible, as only racemic mixtures can be obtained through these methods. On the other hand, pure α-ionone can be obtained via the targeted activity of fungal enzymes [28]. D-limonene, an important industrial monoterpene that occurs as the major component of the oil derived from the peels of oranges and lemons, is widely used in biotransformation processes [29]. To date, countless studies have been conducted on the fungal activity of ascomycetes, but there are very few reports on the microbial activity of basidiomycetes [30].

Zorn [31] screened 50 fungal strains as potential agents for the de novo synthesis or the biotransformation of mono-, sesqui-, tri-, or tetraterpenes by focusing on their ability to cleave β,β-carotene. Here, all strains were grown on plates containing β,β-carotene, and an indication of the biotransformation process was noted as the occurrence of discolouration. Other transformation strategies focused on identifying volatile compounds in the submerged cultures of various basidiomycetes [11,21]. Although the extensive diversity of fungi constitutes a biotechnological potential in Colombia, research in this field is still scarce [32]. Important advances have been made in the description, characterisation, and evaluation of the biological activity of moulds and, more recently, endophytic fungi [33]. However, there are few reports in which the transformation capabilities of macrofungi are examined [34,35,36]. The enzymatic capability of wood-native decay fungi was studied for the degradation of industrial dyes [37,38] and the production of medically important compounds [39].

We believe that, in countries with high biodiversity, the exhaustive search for new active fungal molecules should be focused on poorly studied groups, such as macrofungi like Ascomycota and Basidiomycota, as these organisms may be the key to synthesising new agrochemical and pharmaceutical products. In this study, the biocatalytic potential of native basidiomycetes was examined via the biotransformation of reference molecules—namely, a monoterpene and carotene. Additionally, evidence for the de novo synthesis of volatile aromatic compounds was obtained.

## 2. Results and Discussion

### 2.1. Fungal Strains

Seven strains of Basidiomycota were obtained for this study (Table 1). Two strains belonged to the species *Trametes elegans*; one strain was isolated from a basidioma, and the other was obtained from the endophytic of *Otoba gracilipes*. The molecular identification of each strain at the species level was conducted via amplification, sequencing, and subsequent analysis using the ITS1 region of the rDNA. The sequences generated in this study were deposited in GenBank, and the associated accession numbers are indicated (Table 1). Five of the aforementioned strains belonged to the Polyporaceae family, and the other strain was a member of the Laetiporaceae family. For all datasets, *Ganoderma gibbosum*, *Ganoderma weberianum,* and *Laetiporus gilbertsonii* were new reports in Colombia [40]. Of the Basidiomycota members in this study, polyporales are regarded as vital sources of metabolite-producing fungi [41,42]. In the last decade, this family has become the focus of research efforts in the search for new strains with biotechnological potential [43,44].

In this work, the endophytic basidiomycete *Trametes elegans* was examined; the strain was isolated from the stem of *Otoba gracilipes*, in a tropical montane rainforest in Colombia [45]. The endophytic origin of this species was first reported by Liu et al. [46] after its isolation from the leaves of *Momum villosum* in China. This species was also active against the pathogen *Colletotrichum musae*.

Acronyms for the herbaria are used per the guidelines established in the Index Herbariorum [47] at the Collection of Microorganisms from the Microbiology School of the University of Antioquia, Medellín, Colombia (CM-UDEA).

### 2.2. Screening Basidiomycetes via the Oxidation of β,β-Carotene

All seven strains reported in this work were used to assess the respective microbe’s ability to cleave β,β-carotene, which served as an indicator of biocatalytic activity and the microbe’s potential for producing volatile compounds. The strains were cultivated in nutritive agar containing β,β-carotene. After the fourth day of incubation, *G. stipitatum, G. chocoense*, and *G. webenarium* exhibited a discoloured halo around their respective mycelium (Figure 1). Additionally, general discolouration of the growth medium was observed. For *G. gibbosum* and *T. elegans* (EBB-046), no bleaching effect was observed during the first days of the incubation period. However, considerable discolouration of the agar was noted after 14 days (Figure 1 and Table 2), indicating that the rate of β,β-carotene biotransformation for each fungus was different. We noted that this type of fungus generally exhibited lower growth rates than other filamentous fungi, such as moulds or yeasts. Even though *L. sulphureus* and *T. elegans* (ET-06) did not transform β,β-carotene, they were notably aromatic, as were the other strains. From these findings, we theorised that five of the isolates in our study could be employed for the de novo synthesis of aroma agents via various catalytic mechanisms.

The *Trametes* species is known to produce volatile compounds using multiple strategies. Examples of this can be seen in the fruiting bodies of *Trametes suaveolens*, which produce volatile compounds such as methyl anisate [21]. Zorn reported that, even though both *Trametes suaveolens* and a strain of *Trametes versicolor* were capable of degrading β,β-carotene in whole-cell reactions [31], only *T. versicolor* was capable of producing volatile degradation products such as β-ionone. In the same study, a strain of *Ganoderma applanatum* was only capable of partially transforming β,β-carotene. Even though the mechanism governing the transformation or degradation of β,β-carotene is still unclear, many have attributed the transformation process to the enzymatic activity of carotenoid-cleaving dioxygenases and lipoxygenases (LOX) [20,28]. Peroxidases synthesised by *Lepista irina* and *Mycetinis scorodonius* have been reported as being responsible for the cleavage of this compound [18,48]. The endophytic strain *Trametes elegans* (ET-06) was previously evaluated for its enzymatic LOX activity at various growth phases in submerged cultures [45].

The catalytic proclivity of *G. stipitatum, G. chocoense*, and *G. webenarium* to transform β,β-carotene showed that these microbes were attractive candidates for further screening to determine the enzymes responsible for the aforementioned molecular changes and the formation of volatile flavour compounds.

### 2.3. Biotransformation of α-Pinene

Two strains of *T. elegans* (ET-06 and EBB-046) were shown to biotransform α-pinene into the desired aromatic compound, *cis*-Verbenol. Several factors were taken into consideration when conducting the biotransformation of α-pinene using pregrown whole cells of *T. elegans*. Firstly, ethanol was added as the cosolvent to enhance the interaction between the cellular suspension and α-pinene, thereby reducing the differences in polarity and improving the molecule’s water solubility (0.026 mmol/L) [49]. Secondly, only a small volume of ethanol was used, because high concentrations were known to inhibit the growth of some fungi [50,51]. Lastly, the temperature of the biotransformation reaction was maintained at 22 °C, because the rate of terpene transformation generally increased when the reaction temperature was much lower than the optimal growth temperature of the microorganism. Thus, the oxidation of α-pinene and its conversion to other products of interest was favoured [52]. Additionally, low-temperature biotransformation is particularly beneficial for extremely volatile compounds such as *cis*-Verbenol [25].

The oxidation of α-pinene via microbial transformation was examined, with *cis*-Verbenol, *trans*-Verbenol, verbenone, *trans*-pinocarveol, and mirtenol as the main products [53]. However, only the production of *cis*-Verbenol was analysed in this project, because this precursor was the only one obtained in significant quantities when subjected to biocatalysis using the monooxygenase enzymes found in the cells of various fungal and bacterial species [17,54,55]. Other compounds, such as verbenone and *trans*-Verbenol, were direct products of the autoxidation of α-pinene [56,57]. Table 2 shows the concentration of *cis*-Verbenol obtained from the biotransformation of α-pinene, which was catalysed using the two aforementioned fungal strains (ET-06 and EBB-046) of *T. elegans* at two cell concentrations (i.e., the dry weight of the mycelium, dwm/L)

Independent of the concentration of the mycelium, we noted that relatively higher yields of cis-Verbenol were obtained in the reactions catalysed by the cells of T. elegans (EBB-046) when compared to the results obtained using the endophytic cells (ET-06) (Table 2). Thus, we theorised that the strain of the microbe was the most significant factor influencing the biocatalytic transformations described in this study (*p* = 0.016). Additionally, the cell-free experiments produced results akin to those obtained for the whole-cell oxidation of α-pinene. This could be explained by the cytochrome P450 monooxygenases often found in many species of basidiomycete [55,56,57]. This enzyme oxidises α-pinene in the C4 position, generating cis-Verbenol as the main product (Figure 2). Since these enzymes needed NAD(P)H as a cofactor to carry out this reaction, fungal cells must possess high metabolic activity during biotransformation [58].

The mycelial concentration of *T. elegans* did not significantly affect the biotransformation of α-Pinene to *cis*-Verbenol (*p* > 0.5). The final concentration of *cis*-Verbenol produced by the EBB-046 strain was twice as high as the concentration obtained from the ET-06 endophytic strain (Figure 3). This difference may be due to the biological nature and origin of each strain, since they were collected in different regions and ecosystems. This phenomenon could be an essential factor in determining the metabolism and catalytic activity of any microorganism. Both the fungal response to the media and the enzyme production were regulated by catabolite repression, as evidenced by the EBB-06 strain, which was isolated from basidioma growing in the trunk of a tree. We theorised that these white-rot fungi must have an active decomposition mechanism compared to that of the endophytic strain. Conversely, endophytic fungi tended to form complex symbiotic relationships with their plant hosts, presumably receiving carbon-containing substances such as sugars for nourishment. In this case, its metabolism was different from that of the other specimens that served as agents of degradation [59]. Both strains have vastly different genetic materials. Therefore, the expressions of different genes in each strain was quite notable, particularly the genes related to enzyme production, including cytochrome P450 monooxygenase [60]. A report on unidentified fungal endophytes isolated from Baru (*Dipteryx alata Vog.*) in Brazil showed that these microbes transformed α-pinene into *cis*-Verbenol [61]. However, there was no clear information about the reaction conditions, particularly data regarding the concentration of the biomass used in the aforementioned trials.

We noted, rather unexpectedly, that a low concentration of *cis*-Verbenol was obtained after the biotransformation of 20 mM of α-pinene catalysed by a cellular concentration of 10 g/L of EBB-046. This finding was attributed to the mechanism governing α-pinene′s oxidation, since *cis*-Verbenol was not the final product of this reaction, but rather, it was an intermediate or precursor. Krings [57] reported that *cis*-Verbenol was oxidised and transformed into verbenone when there were high concentrations of dehydrogenase in the reaction mixture. In our experiment, a considerable fraction of the *cis*-Verbenol product could be converted into verbenone due to the high catalytic potential of EBB-046 at a mycelial concentration of 10 g/L. This was associated with a high risk of toxicity that tended to cause the catalytic inhibition of specific reactions in many microorganisms [62] or were due to a decrease in its oxygenase activity and its deficiency as a cofactor [25].

The concentration of the initial substrate (α-pinene) for subsequent biotransformation into *cis*-Verbenol was varied depending on the strain of *T. elegans* used in the respective experiments. A correlation was observed in Table 2 between the initial concentration of α-Pinene and the *cis*-Verbenol product formed via the biocatalytic action of ET-06. A higher concentration of *cis*-Verbenol was obtained by using the highest amount of the initial substrate (i.e., 20 mM). This correlation could be explained by the mechanism governing the reaction between the substrate (α-pinene) and cytochrome P450 monooxygenase. Enzyme-catalysed reactions that follow the Michaelis–Menten kinetic model generally exhibited faster reaction rates beyond the saturation zone as a function of the substrate’s concentration; this was because more substrate molecules interacted faster with the catalyst to form the E–S complex [63]. For the strain EBB-046 using 10-g/L cells, a lower *cis*-Verbenol yield was obtained when 20 mM of α-pinene was used as the initial substrate. Given these results, we theorised that the initial concentration of the substrate did not have a statistically significant effect (*p* = 0.339).

Finally, the efficiency of the applied techniques and procedures was determined by quantifying the percentage of *cis*-Verbenol recovered from the biotransformation and extraction processes. Here, an initial quantity of *cis*-Verbenol (0.15 mM) was added to the system. Extraction and separation procedures were performed to quantify the final concentration of *cis*-Verbenol obtained after a 48 h incubation period, which was noted at 0.061 mM of *cis*-Verbenol; this value corresponded to 40.7% of the initial metabolite. At first glance, it could be said that the efficiency of this method was very low. However, this statement was not entirely correct, since *cis*-Verbenol was known to be unstable when there were drastic changes in the temperature and acidity. In these cases, *cis*-Verbenol tended to change its atomic distribution and adopted the configuration of its *trans*-counterpart [64]. This molecule was an important precursor of verbenone when it was catalysed by the dehydrogenase enzymes found in several basidiomycete fungi [57]. Therefore, we inferred that the 48-h incubation period was sufficiently long to allow for the oxidative transformation of a large percentage of the initial *cis*-Verbenol product into other compounds such as verbenone and *trans*-Verbenol.

In the biotransformation reaction catalysed by the cells of the *T. elegans* strains, a higher concentration of *cis*-Verbenol (0.1 mM) was obtained than that observed in the experiments without the cells (control) (0.03 mM). This was indicative of the catalytic action of an enzyme such as cytochrome P450 monooxygenase, which has been reported in various basidiomycete species such as *Pleurotus sapidus* and *Pleurotus eryngii* [57].

Figure 3 shows the ratio of the *cis*-Verbenol yield relative to the total mycelial mass (g) used for the whole-cell biotransformation reactions with a 48-h incubation period for both *T. elegans* strains. Here, the amount of product recovered was an indication of the total amount of the biocatalyst added to the reaction. The yields for the endophyte, which were around 1.5-mg *cis*-Verbenol per g of the dry weight mass (dwm), were very similar to that observed for the other strain and appeared to be unaffected by the concentration of α-pinene and the mycelia; the EBB-046 strain produced a two-fold higher yield when the concentration of α-pinene was 10 mM and the mycelial level was at its lowest (5 g/L). In our study, the proportion (v/v) of the substrate (α-pinene) to the media was kept at 0.032%, a value that was much lower than the 0.6% (*v*/*v*) reported by Vespermann [17] for biotransformations using *Aspergillus niger.* In general, we noted lower product yields at higher substrate concentrations, longer reaction times, or higher temperatures (> 28.5 °C).

Herein, we noted that this behaviour was not necessarily the same for all the basidiomycetes tested. In particular, we found fundamental differences between both strains of *T. elegans.* For example, the biotransformation of α-pinene using pregrown mycelia at 20 °C and only a fraction of the substrate (i.e., 0.3 *v*/*v*) yielded values less than 1 (g *cis*-Verbenol/g dwm L) when the fungus *Chrysosporium pannorum* was used [25]. This value was comparatively lower than the value obtained from the EBB-046 strain using a shorter incubation time (i.e., 48 h). This difference indicated that our strain was a markedly better biocatalyst for the biotransformation of this terpenoid.

### 2.4. Identifying the Volatile Products of the Trametes Elegans Strains in Submerged Cultures

#### 2.4.1. Submerged Culture

In a preliminary test, two different morphologies were obtained for *T. elegans* (EBB-046) during the development of the inoculum using the yeast extract medium (Figure 4A,B). The clumpy morphology (Figure 4B) was characterised by pleasant, very intense aromas relative to their mycelial counterparts that adopted a pellet-like form (Figure 4A). For this reason, the clumped mycelia were used as the pre-inoculum for submerged culture experiments in an agitated tank bioreactor with a medium formulated using Cape gooseberry as the only carbon source. Previous studies showed that the characteristic morphological regulation processes of the ligninolytic enzymes obtained from the other species of *Trametes*, in which the highest enzyme production levels were achieved, resulted in a pellet-like structure during the submerged culture experiments [65]; this was contrary to the effect observed with both strains of *Trametes elegans* in this work. From this finding, it was clear that the inoculum′s condition played a pivotal role in both the regulation of the metabolites (i.e., enzymes and VOCs) and the activation of this fungal genus.

The appearance of the submerged culture after 96 h is shown in Figure 4. Here, the observed homogeneous turbidity was attributable to variations in the initial morphology of the mycelia from clumps to the free structures. This morphological change possibly occurred due to aeration and agitation during the culture’s incubation, which affected the mechanisms governing mycelial aggregation. Additionally, this strain was capable of using the sugars derived from Cape gooseberry as carbon sources for growth. Previous studies have reported that the biotransformation of apple pomace by various basidiomycetes, including *Trametes suaveolens* and *Trametes versicolor*, resulted in flavour agents or aromatics [66].

#### 2.4.2. Sensory Evaluation

Figure 5 shows the incremental progression of the intensity of the pleasant aromas (i.e., the increase in sweet and citrus fruit notes) generated by the biotransformation reaction in the Cape Gooseberry medium via the *Trametes elegans* mycelia. No unpleasant flavours were noted during the sampling period, as described by the expert panel. These variations in intensity could be linked to changes in the metabolites’ concentration in response to the fungal metabolism. One example of this was noted in the basidiomycete *Nidula niveo-tomentosa*, which was used to synthesise a raspberry-scented ketone via submerged cultivation with a 50-fold increase in the product yield at the end of the incubation period [67]. The resulting odour obtained from a strain of *Trametes pini* cultivated via submerged culture experiments both with agitation and static (without mixing) was described as “fruity” [68]. Similarly, the media composition was supplemented with amino acids that served as flavour activators.

Even though the positive response of this strain produced flavour/aroma incrementally under the culture conditions stated above, further evaluations are needed to obtain extensive details about the microbe’s usefulness for strategic biotechnological applications.

#### 2.4.3. Gas Chromatography

The volatile compounds obtained from the culture broth of *Trametes elegans* (EBB-046) via the submerged cultures were identified using gas chromatography–mass spectroscopy (GC–MS). The resulting total ion chromatogram for the volatile compounds is shown in Figure 6. Here, presumptive identification was based on the mass spectrum obtained at an electron ionization (EI) of 70 eV using the Adams, Wiley, and National Institute of Standards and Technology (NIST). A total of eight organic compounds were identified (Table 3). The first three peaks were present in the negative control. The second one, which corresponded to maltol, was formed during the thermal treatment (i.e., sterilisation) of the Cape gooseberry substrate. This potent flavour enhancer, which is typically added to foods and beverages, can undergo thermal degradation to produce polysaccharides [69].

Only three molecules (i.e., peaks 4, 5, and 6) and their associated isomers were attributed to the sweet/fruity aroma of the broth and were not detected in the control samples. The corresponding mass spectra, in which these compounds were identified as heterocyclic ketones, can be found in the Appendix A.

Headspace gas chromatography/mass spectrometric (HS-GC/MS) techniques have been used to quantify volatile organic and flavour compounds from several microorganisms. In one study, the aromatic compound 2,5-dihydro-3,5-dimethyl-2-furanone was identified as a ketone derived from the mycelium of *Ganoderma sinense* [10]. Recently, the basidiomycete *Pleurotus flabellatus* was used to transform the hairy roots of *Hypericum perforatum*, resulting in several odour compounds that exhibited a distinct fruity aroma of average intensity attributable to 1-methoxy-4-methylbenzene [2]. The white-rot *Trametes versicolor* produce VOCs as a function of specific reactions championed by other fungal species, resulting in monoterpenes, alkenols, and quinolinium-like compounds [70].

Similar heterocyclic molecules—namely, 2,5-pyrrolodonedione and 7-benzylidene-6(biphenyl-2-yl)-6,7-dihydropyrrolo[3,4-b]pyridine-5-one—were bioactive compounds obtained from the fresh and dried biomass of *Pleurotus ostreatus* [71]. The fruity note assessed at the beginning of the culture (Figure 5) corresponded to the enhancer effect exerted by maltol on the original notes from the Cape gooseberry substrate. However, the compounds responsible for this aroma—namely, methyl-2-methyl butanoate, methyl-3-methyl butanoate, methyl hexanoate, methyl benzoate (trace amounts), and methyl salicylate [72]—were not found in the analysis. This observation indicated that the initial notes were from the inoculum broth and, thus, could be synthesised by our evaluated strain.

Further studies using submerged cultivations of this native strain with various natural substrates (i.e., tropical fruits and their by-products) in the medium must be conducted to gain extensive knowledge about a diverse collection of flavour/aroma molecules. Herein, the main focus of this work was to use environmentally friendly extraction methods that required no organic solvents. Note, however, that the resulting spectra obtained for the isolated molecules depended on the target application.

## 3. Materials and Methods

### 3.1. Isolating the Microorganisms and Conducting Molecular Identification

Basidiomycetes strains were obtained from basidiomas collected in the tropical forests of Colombia in Antioquia and Valle del Cauca. The strains of the endophytic fungus were isolated from the leaves of *Otoba gracilipes* (Myristicaceae) that were collected in Valle del Cauca [73]. Briefly, sections of the fresh basidiomas were cleaned with a series of sequential washing steps for one min each as follows: Tween 80 (0.01%), sterile water, 70% ethanol, sterile water, 70% ethanol, and lastly, another round of sterile water. The sections were seeded in plates with 2% water–agar or half-potato dextrose agar (signal PDA) containing 0.01% chloramphenicol [74]. The strains obtained were preserved in 20% glycerol (*v*/*v*) at −20 °C, and samples of each strain were deposited in the microorganism collection of Universidad Icesi or the Microorganism Collection, School of Microbiology, UDEA (CM-EM-UDEA).

Molecular identification was conducted using the strains that were previously grown on PDA and incubated for 5–7 days at 29 °C. The fungal DNA was extracted using an EZNA^®^ Tissue DNA Kit (Omega Bio-Tek, Norcross, GA, USA), and the complete DNA profile was quantified using a NanoDrop spectrophotometer 2000/2000c ND-1000 (NanoDrop, Wilmington, DE, USA). The nuclear ribosomal ITS1 region was amplified with the primers ITS 1 and ITS 5 [75]. DNA amplification was performed in a Swift™ MiniPro Thermal Cycler (ESCO, Singapore) with an initial denaturation step for 1 min at 95 °C, followed by 35 cycles of denaturation for 1 min at 95 °C, annealing for 30 sec at 52 °C, and an extension for 30 sec at 72 °C. A final extension at 72 °C was performed for 5 min. The PCR products were visualised using 1% agarose gel. Purification of the products was conducted using Wizard SV Gel and PCR Clean-Up System (Promega, San Luis Obispo, CA, USA) before being subjected to sequencing protocols using the Applied Biosystems^®^ ABI prism 3500 sequencer (Thermo Fisher Scientific, Waltham, MA, USA). The resulting DNA sequences were analysed and compared with those obtained from GenBank via a BLAST search [76]. The sequences from our study were also deposited in GenBank.

All the protocols and procedures employed in this investigation were verified and approved by the appropriate institutional review committee. The specimens were kept and handled in accord with the guidelines of the “Autoridad Nacional de Licencias Ambientales (ANLA)-Colombia” through permission “Permiso Marco de Recolección de Especímenes de Especies Silvestres de la Diversidad Biológica con Fines de Investigación Científica No Comercial”—Resolution 0526, 20 May 2016. Furthermore, according to Resolution 0364, 12 March 2018 and otrosí No. 4 del Contrato Marco de Acceso a Recursos Genéticos y sus Productos Derivados No. 180 de 2018, Universidad Icesi has the acceptance to the request of “Contrato Marco de Acceso a Recursos Genéticos y sus Productos Derivados” for “Programa para el Estudio, Uso y Aprovechamiento Sostenible de la Biodiversidad Colombiana”.

### 3.2. Screening the Catalytic Action of Basidiomycetes Using β,β-Carotene Oxidation

All seven fungal strains were grown on Petri dishes containing nutritive agar and β,β-carotene (≥93%; Sigma–Aldrich, St. Louis, MO, USA), with a final concentration of 0.025 g/L per the method reported by Zorn [31]. Here, the carotene was emulsified with the media using Tween 80 and dichloromethane as the solvent. The Petri dishes were covered with aluminium foil and incubated at 28 °C with continuous monitoring for 14 days. The biotransformation of β,β-carotene was observed as decolourisation of the areas surrounding the mycelia.

### 3.3. Biocatalytic Transformation of α-Pinene

#### 3.3.1. Inoculum Development

The ET-06 and EBB-046 strains of *T. elegans* were maintained on PDA slants stored at 4 °C and subcultured monthly. The cells from the conserved strains stored at −20 °C were activated on the PDA plates at 28 °C for 7 days. Next, ten 0.5-cm agar fragments containing fungal mycelia were removed and inoculated in 50 mL of liquid Yeast extract-Malt extract medium (YM). After a 5-day incubation period, the new biomass, which exhibited a clumpy appearance, was used as the inoculum for future cultures.

#### 3.3.2. Producing the Mycelial Biomass for Biotransformation Reactions

The pregrown mycelia obtained from the inoculum, each of which was grown in 500 mL of the YM medium with stirring in a tank reactor (Applikon Biotechnology, Delft, The Netherlands) for 7 days, were used for further experimentation. The controlled culture conditions were 28 °C, pH 5.5, 100 rpm, and 1.0 *v*/*v*/*m*. The fermentation was performed for a week with daily sampling to quantify the biomass via gravimetry. At the end of this timeframe (i.e., the beginning of the stationary phase), the entire culture medium was centrifuged for 10 min at 5000 rpm to recover the biomass for subsequent α-pinene biocatalytic transformation assays.

#### 3.3.3. Biocatalytic Transformation Reactions of α-Pinene using the Trametes Elegans Mycelia

The experimental design featured the respective fungal strain of *T. elegans* (i.e., ET-06 and EEB-046), the mycelial concentration (5 and 10 g of dry biomass per L), and the α-pinene concentration (10 and 20 mM). All experiments were performed in duplicate, and the standard deviations for the corresponding experiments were determined.

For the bioconversion of α-pinene, 4840 μL of the biomass was suspended in a phosphate buffer (pH 5.5 and 0.5 M), and 160 μL of an α-pinene solution (10% *v*/*v* in ethanol) was added. Ethanol was used as the cosolvent to improve the substrate’s solubility during biotransformation, as reported by Rojas [51]. The reactions were performed in 15-mL glass test tubes and covered with aluminium foil. These tubes were subjected to orbital agitation (150 rpm) at a constant temperature of 22 °C on a Heidolph^TM^ Unimax 1010 shaker (Schwabach, Germany). After 48 h, 1.5 mL of an ethyl acetate/hexane (1:1 *v*/*v*) mixture was added, followed by centrifugation at 4000 rpm for 15 min. The supernatant (i.e., the upper layer) was recovered and subjected to extraction protocols to obtain the resulting metabolites [51].

#### 3.3.4. Quantitative Analysis of the cis-Verbenol Content via GC

The solvent was removed under vacuum to yield 1 mL of the extract, to which gaseous nitrogen and *n*-tetradecane (78.5 uL) as the internal standard were added. Sample analysis was performed on an Agilent 6890N gas chromatograph (Agilent Technologies, Palo Alto, CA, USA) with a flame ionisation detector. The chromatography column used for purification was a D8-5MS column (J&W Scientific, Folsom, CA, USA) composed of 5% phenylpoly (dimethylsiloxane) (60 m × 0.25 mm × 0.25 µm). The injection was performed in the split mode (30:1), with an injection volume of 2µL [51]. (1S)-(−)-α-Pinene (98%), (S)-*cis*-verbenol (95%), and *n*-tetradecane (99%) were purchased from Sigma–Aldrich, USA.

### 3.4. Identifying Volatile Compounds Derived from Trametes Elegans (EBB-046) in Submerged Cultures

#### 3.4.1. Submerged Culture

Substrate preparation: Cape gooseberry (*Physalis peruviana* L.), a plant commonly found in the South American Andes, is characterised by sugary fruits with a high content of bioactive compounds such as ascorbic acid (Vitamin C), β-carotene (provitamin A), phenols, and carbohydrates [77]. The fruit’s pulp was used as the substrate for the submerged culture. The fruit’s pulp was homogenised, and a solution was subsequently prepared and sterilised by autoclaving. The final concentration of the solids was 70 g/L.

Precultures and fermentation of the substrate: The clumpy inoculum was added to the medium containing urea (0.65 g/L), (NH_4_)_2_Cl(1.15 g/L), L-phenylalanine as the inductor (1.0 g/L), CaCl_2_.2H_2_O (0.013 g/L), MgSO_4_.7H_2_O (0.5 g/L), and the substrate solution (500 mL). The pH of the medium was adjusted to 5.5 by adding sodium hydroxide (0.5 M) or hydrogen chloride (0.5 M) before conducting the fermentation experiments. The medium was autoclaved for 20 min at 121 °C. The bioreactor (Sartorius Stedim Biotech, Göttingen, Germany) with 3.5 litres of the medium was started at 28 °C, 65 rpm, 0.2 *v/v/m*, and pH 5.5. The fermentation was conducted for 90 h, during which samples were taken at different time intervals and analysed via sensory evaluation.

#### 3.4.2. Sensory Evaluation of the Culture

Liquid aliquots (25 mL) containing the supernatant and mycelia were taken during the fermentation process and transferred to odourless cups (50 mL). The sample was then placed in a convection oven (BINDER GmbH, Crailsheim, Germany) for 20 min at 80 °C. The sensory evaluation was conducted by three expert panellists to establish the odour impression, which contained the description of the aroma and its associated intensity. The odour intensity was rated on a scale of one to eight, in which 1–2 represented “low intensity”, 3–4 was noted as “characteristic”, 5–6 was “intense”, and 7–8 represented “strong”. When the smell was unpleasant or dissimilar from the target scent, which was characterised by a pleasant complex natural sweet/fruity flavour, a negative intensity value was given by the experts.

#### 3.4.3. Gas Chromatography

Numerous samples (10 mL) from the submerged culture at the end of the cultivation period were transferred into a Büchi R100 rotavapor system (Büchi Labortechnik AG, Flawil, Switzerland) and distilled for 15 min at 40 °C at 55 mBar [78]. In this case, solvent extraction was avoided to ensure a narrower spectrum of analytes that were beyond the scope of our research. All fractions were collected in 2-mL amber flasks for subsequent analysis and the identification of the volatile organic compounds in the culture broth of *Trametes elegans* (EBB-046). The negative control of the media, i.e., the cell-free media, was subjected to the same conditions. The extracted compounds were identified using a coupled GC–MS system to determine the molecular weight of the products and ions formed via fragmentation. The analysis was performed on an AT6890 Series Plus gas chromatograph (Agilent Technologies, Palo Alto, CA, USA) coupled to a mass-selective detector (Agilent Technologies, MSD 5975) operating in the full-scan mode. The column used in the analysis was a 5%-phenylpoly (dimethylsiloxane) DB-5MS (J&W Scientific, Folsom, CA, USA) with the dimensions of 60 m × 0.25 mm × 0.25 µm.

The identification of compounds was made by comparison of experimental mass spectra with those from databases (NIST and Wiley) and using a fragmentation pattern study. The confirmation requires the use of standard compounds.

## 4. Conclusions

Through the application of several screening methods, the incredible biocatalytic potential of native strains isolated from Colombian plants was examined for the biotechnological production of flavour/aroma alternatives. Herein, the compounds were generated via whole-cell biotransformation or de novo biosynthesis. Even though the screening for the new basidiomycetes to determine their biocatalytic capabilities for the transformation of β,β-carotene was relatively easy, this method of quantification alone was insufficient. In our case, even when endophytic *T. elegans* and *L. sulphureus* were the only strains incapable of biotransforming β,β-carotene, these microbes generated a characteristic, pleasant aroma. The rest of the strains examined in this study exhibited discolouration during the first days of incubation, with *G. webenarium, G. chocoense*, and *G. stipitatum* being listed as the most efficient biotransformation agents. Additionally, the strain that exhibited moderate activity—namely, *Trametes elegans* that were isolated from a basidioma—produced aromatic compounds and flavour agents via de novo synthesis. We noted that, even though some strains appeared to be possible agents for producing the desired aromatic compounds from other substrates, they were incapable of degrading the terpenoid α-pinene. Further experiments via liquid cultivation techniques must be performed on the aforementioned strains to determine the type of molecules that could be synthesised by these fungi via the degradation of β,β-carotene.

The biocatalytic potential to produce aromatic compounds notably varied depending on the origin of the strain. The strains of *T. elegans* (i.e., from basidioma and endophytic) evaluated in this study showed different biocatalytic capabilities toward β,β-carotene, yet both transformed α-pinene through whole-cell biotransformation reactions with varying efficiency. Herein is the first reported application of this species for the biocatalytic oxidation of α-pinene. The strain isolated from a basidioma—namely, EBB-046—presented a higher biocatalytic potential for α-pinene oxidation. However, both species exhibited better biotransformation performances when compared to the results of other studies. This study demonstrated that an efficient primary screening methodology for evaluating basidiomycetes must entail the integration of various analytical approaches to gain a comprehensive understanding of the biocatalytic potential of these microorganisms.

## Figures and Tables

**Figure 1 molecules-25-04344-f001:**
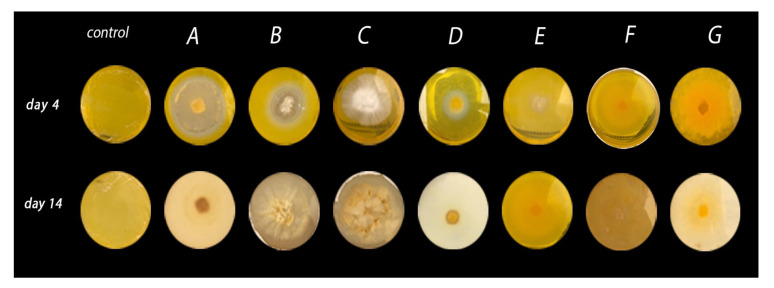
β,β-carotene biotransformation after a 4- and 14-day incubation periods. Screening was conducted using seven strains of native basidiomycetes—namely, *Ganoderma stipitatum* (**A**), *Ganoderma chocoense* (**B**), *Ganoderma gibbosum* (**C**), *Ganoderma weberianum* (**D**), *Laetiporus gilbertsonii* (**E**), *Trametes elegans* (endophyte) (**F**), and *Trametes elegans* (**G**).

**Figure 2 molecules-25-04344-f002:**
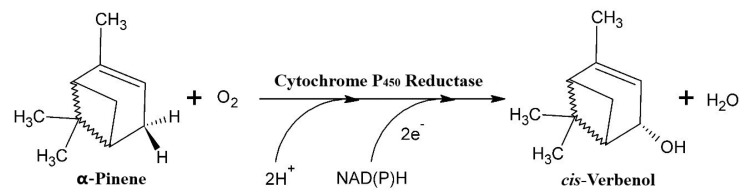
The hydroxylation of α-pinene via the P450 monooxygenase/reductase system (adapted from Krings [57]).

**Figure 3 molecules-25-04344-f003:**
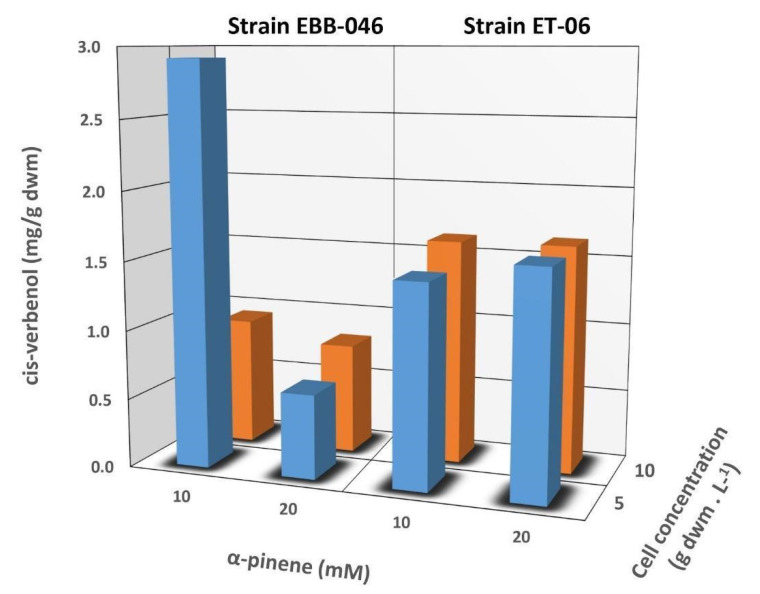
The influence exerted by the concentration of α-pinene (10 and 20 mM) and the mycelium (5 and 10 g/L) on the *cis*-Verbenol yield during biotransformation reactions using whole cells of the EBB-046 and ET-06 strains of *Trametes elegans*. The incubation time was 48 h.

**Figure 4 molecules-25-04344-f004:**
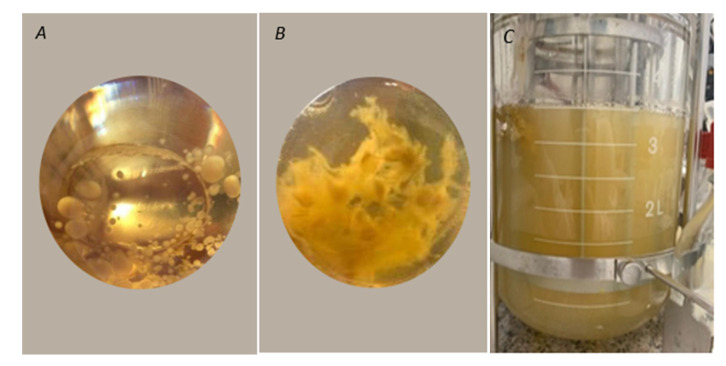
(**A**) Pellets formed by the EBB-046 strain in the Yeast extract-Malt extract medium YM. (**B**) Mycelial clumps formed by the EBB-046 strain in the yeast extract medium. (**C**) Image of the submerged culture of *Trametes elegans* (EBB-046 strain) in a defined medium using Cape Gooseberry after an incubation period of 96 h.

**Figure 5 molecules-25-04344-f005:**
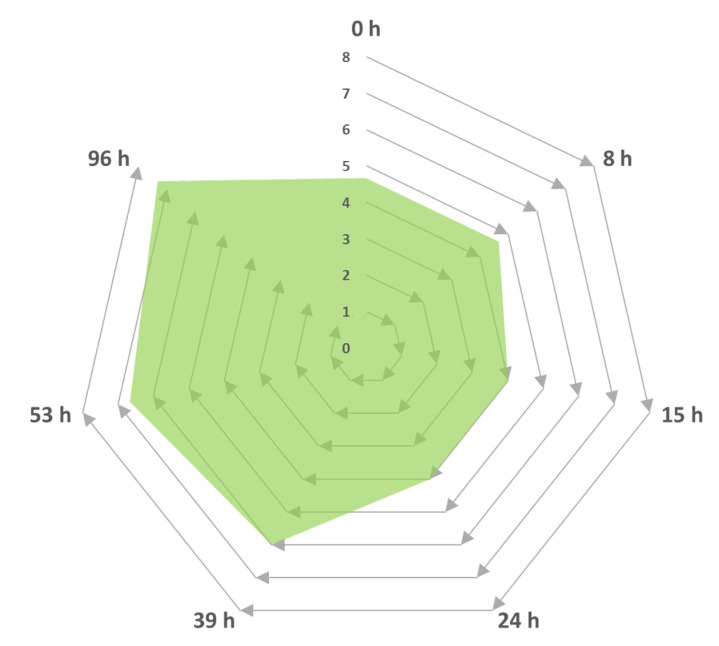
Intensity progress descriptor for the generation of sweet and fruity odours as a function of time by *Trametes elegans* mycelia cultivated in media containing Cape Gooseberry. The odour intensity was rated on a scale of one to eight, where 1–2 represented a low intensity, 3–4 was described as “characteristic”, 5–6 was classified as “intense”, and 7–8 was a strong odour.

**Figure 6 molecules-25-04344-f006:**
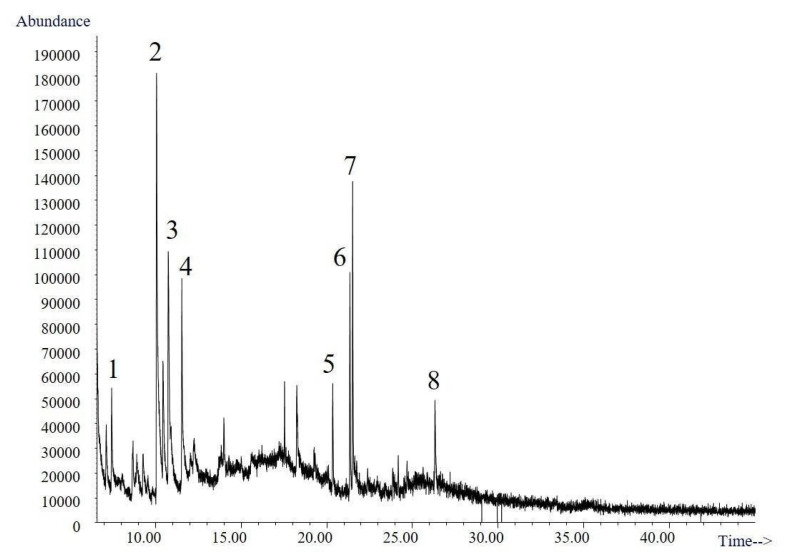
Chromatogram of the biotransformation products of *Trametes elegans* (EBB-046) obtained after an 80-h incubation period.

**Table 1 molecules-25-04344-t001:** List of fungal strains used in this study.

Taxa	Strain	Voucher Information *	GenBank Accession Numbers	Degradation Ability *
*Ganoderma chocoense* J.A. Flores, C.W. Barnes & Ordoñez	EBB-083	EBB-083 (Icesi); Colombia, Valle del Cauca, Reserva La Carolina	MT945608	++
*Ganoderma gibbosum* (Cooke) Pat.	CM-UDEA 10	2543 AMV (HUA); Colombia, Antioquia, Medellin, on trunk	MT945606	+
*Ganoderma stipitatum* (Murrill) Murril	CM-UDEA 110	2526 AMV (HUA); Colombia, Antioquia, San Luis, Reserva Río Claro	MT945605	++
*Ganoderma weberianum* (Bres. & Henn. ex Sacc.) Steyaert	CM-UDEA 111	2531 AMV (HUA); Colombia, Antioquia, San Luis, Reserva Río Claro	MT953467	++
*Laetiporus gilbertsonii* Burds.	CM-UDEA 1	2483 AMV (HUA); Colombia, Antioquia, Medellin, on trunk	MT945604	-
*Trametes elegans* (Spreng.) Fr.	ET-06	Endophytic isolated from Otoba gracilipes (Myristicaceae)	MT941002	-
*Trametes elegans* (Spreng.) Fr.	EBB-046	2472 AMV (HUA); Colombia, Antioquia, Santafé de Antioquia on trunk	MT945607	+

* Basidiomycetes biotransformed β,β-carotene when grown on β,β-carotene-containing agar plates.^+^ Pronounced bleaching of the agar was noted after a 14-day incubation period.^++^ Pronounced bleaching zone around the mycelium was noted after a 4-day incubation period.- No colour fading was noted after incubation periods of 4 or 14 days*. CM-UDEA: Collection of Microorganisms from the Microbiology School of the University of Antioquia.

**Table 2 molecules-25-04344-t002:** Concentration of *cis*-Verbenol (mM) noted during the biotransformation of α-pinene using the whole cells of *T. elegans* (ET-06 and EBB-046) in a 48-h reaction period. dwm: dry weight mass.

Cell Concentration (g dwm L^−1^)	Concentration of *cis*-Verbenol (mM)			
	*T. elegan*s ET-06		*T. elegan*s EBB-046	
	α-pinene 10 mM	α-pinene 20 mM	α-pinene 10 mM	α-pinene 20 mM
5	0.048 ± 0.011	0.086 ± 0.001	0.107 ± 0.002	0.103 ± 0.038
10	0.052 ± 0.001	0.095 ± 0.004	0.107 ± 0.011	0.078 ± 0.019
0 (control)	0.030 ± 0.005	0.037 ± 0.004	0.030 ± 0.005	0.037 ± 0.004

**Table 3 molecules-25-04344-t003:** Biotransformation products of *Trametes elegans* (EBB-046).

N^o^	*t_R_* (min)	Presumptive Identification	MW
1	7.44	2,3-Dihydro-3,5-dihydroxy-methyl-4H-pyran-4-one	144
2	10.13	Maltol	126
3	10.78	2,3-Dihydro-3,5-dihydroxy-methyl- 4H-pyran-4-one (isomer)	144
4	11.54	Methyl-3-methoxy-4H-pyran-4-one	140
5	20.32–20.36	Hexahydro-3-(methylpropyl)-pyrrolo[1,2-C-a]pyrazine-1,4-dione	210
6	21.34	Hexahydro-3-(methylpropyl)-pyrrolo[1,2-C-a]pyrazine-1,4-dione (isomer)	210
7	21.35	Hexahydro-3-(methylpropyl)-pyrrolo[1,2-a]pyrazine-1,4-dione (isomer)	210
8	26.30	Hexahydro-3-(methylphenyl)-pyrrolo[1,2-a]pyrazine-1,4-dione	244

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
