# Peer review of "Biocatalytic Potential of Native Basidiomycetes from Colombia for Flavour/Aroma Production"

_molecules, 2020, doi:10.3390/molecules25184344_

Round 1
Reviewer 1 Report
Interesting manuscript, few English corrections necessary.

Author Response
Dear Reviewer,
the manuscript has been already revised and corrected by the Editor Service of ENAGO:

Reviewer 2 Report
Article: molecules-902568
The present paper reports the evaluation of the biocatalytic potential of native basidiomycetes fungi to biotransform b-carotene in the respective a- and b-ionones and a-pinene in verbenol. This is an interesting work due to its biotechnological approach. However, in my opinion, there are some aspects which must be clarified prior to the acceptance in Molecules
- The authors describe, in different parts of the manuscript, the term b,b-carotene. Is this correct?
- It is not clear to me why the authors decide to test the biotransformation of pinene/carotene. There are several monoterpenes and carotenes reported in the literature. For example, limonene is a product obtained on a large scale in the Citrus industry and could be used in different assays in order to evaluate its biotransformation. This point must be clarified in the revised version.
- b-carotene and a-pinene are both very lipophilic compounds. How these compounds were administered in the fungi medium? The different polarity of medium and tested compounds could not be a limitation to use this approach on a large scale?
- How was determined the higher concentration of pinane/carotene to be biotransformed using tested fungi?
- Please, correct figure 2. The molecule of pinane is inadequately presented.
- Please, include additional details concerning the yield of biotransformation.
- Analysis using GC and GC/MS was performed using crude extracts using the head-space approach. Why these extracts were not extracted using volatile solvents such as hexane, CH2Cl2, or ethyl ether? This protocol is usual and adequate for the extraction of volatile and non-polar compounds.
Author Response
1) It is not clear to me why the authors decide to test the biotransformation of pinene/carotene. There are several monoterpenes and carotenes reported in the literature. For example, limonene is a product obtained on a large scale in the Citrus industry and could be used in different assays in order to evaluate its biotransformation. This point must be clarified in the revised version.
We agree with this comment; however, as it was clarified in the revised version, the limonene is more commonly biotransformed by ascomycetes than by basidiomycetes. This means that α-pinene and β-carotene are more appropriate reference molecules to evaluate the biocatalytic potential of these basidiomycetes.
Pinene is a molecule present in pine-oil, which is commonly used in the aroma industry.
Let us make emphasize that we wanted to fix some reference molecules to be transformed as a systematic platform for the evaluation of the biocatalytic potential of basidiomycetes.
2) The authors describe, in different parts of the manuscript, the term b,b-carotene. Is this correct?
According to IUPAC nomenclature, the term ?,?-carotene is correct. In the literature, both the IUPAC nomenclature and the commercial term which is only ?-carotene are cited. The name according to IUPAC in the 2005 edition, is mentioned (International union of pure and applied chemistry. Division of chemical nomenclature and structure representation. _Favre, Henri A. _Powell, Warren H - Nomenclature of organic chemistry_ IUPAC recommendation-2005) We prefer to keep this nomenclature throughout the text.
3) Please, correct figure 2. The molecule of pinane is inadequately presented.
In the figure, we are presenting the structure of alpha-pinene and not pinane. The chemical structure we used in the paper is correct.
4) Please, include additional details concerning the yield of biotransformation.
This detail has already been added to the revised version.
5) How was determined the higher concentration of pinane/carotene to be biotransformed using tested fungi?
This concentration was not determined because it was out of the scope of the research. However, the used values were following the reported in the literature, and they were always out of the saturation zone.
6) B-carotene and a-pinene are both very lipophilic compounds. How were these compounds administered in the fungi medium? The different polarity of medium and tested compounds could not be a limitation to use this approach on a large scale?
We agree with this comment; however, there are several strategies to improve these phase limitations during the reaction, such as the applied in our case (ethanol for α-pinene and dichloromethane for β-carotene). Even Though this phenomenon can be a limitation for large scale processes, it is important to consider these lipophilic molecules during these screening steps because the broad spectrum of possibilities can be reached by the action of biocatalytic capabilities of fungi. In effect, the biocatalytic performance of many enzymes and whole-cell systems is enormously improved under non-conventional conditions, which includes the use of solvents.
7) Analysis using GC and GC/MS was performed using crude extracts using the head-space approach. Why were these extracts not extracted using volatile solvents such as hexane, CH2Cl2, or ethyl ether? This protocol is usual and adequate for the extraction of volatile and non-polar compounds.
We agree with this comment; however, our approach was more focused on a green extraction process, and for this reason, we wanted to avoid the use of an organic solvent in this step. We have chromatograms of extracts obtained after using ethyl ether as a solvent, which could be attached as evidence if you consider it necessary.

Reviewer 3 Report
Dear all,
below are my comments and suggestions
Manuscript ID: molecules-902568
Title: Biocatalytic potential of native Basidiomycetes from Colombia for flavor/aroma production
In this study, the authors report the possibility of using the fungi, that is an alternative to obtain natural compounds by de novo synthesis and by biotransformation and evaluate the biocatalytic potential of the some new basidiomycetes species.
The novelty of the work is to evidence the de novo synthesis of volatile aromatic compounds, with with potential industrial applications.
Point 1. The abstract should include methods and at least some significant results;
Point 2. Please indicate the international regulations regarding a natural flavor can be obtained via microbial or enzymatic processes,
Point 3. You didn't do a statistical analysis of the results, I recommend you using it in the analysis of the results
Point 4. The number of panelists (3) is very low for obtain a significant results.
Point 5. Section 3. Materials and Methods.
Section 3.4.3. This part is a description of methods, please indicate the bibliographic sources
Point 6. The English language should be revised in all the paper (errors in spelling, grammar, and style)
Point 7. Please check the style for citations and references in all the paper
The number of bibliographic sources is adequate, but less than 35% of the total bibliographic sources are from the last 5 years.
Author Response
Point 1. The abstract should include methods and at least some significant results
We have included some results in this section.
Point 2. Please indicate the international regulations regarding a natural flavor can be obtained via microbial or enzymatic processes,
We specify the regulation used: 5. Regulation (EC) No 1334/2008 of the European Parliament and of the Council of 16 December2008 on flavourings and certain food ingredients with flavouring properties for use in and on foods and amending Council Regulation (EC) No 1601/91. Regulations (EC) No 2232 and (EC) No 110/2008 and Directive 2000/13/EC (Text with EEA relevance) Official Journal of the European Union 34, European Parliament and Council 16,12.2008.
Point 3. You didn't do a statistical analysis of the results; I recommend you using it in the analysis of the results
This aspect was correctly modified in the revised version.
-Point 4.The number of panelists (3) is very low for obtaining a significant result.
You are right in this comment; however, let us inform you that all the panelists belong to a specialist group from the company LaTour, which specializes in the production of aroma/flavors. They are frequently trained for the identification of the notes of interest (fruity, sweet) we checked in our work. Taking into account this, we consider the assessment means, which is shown in the paper, as a good indicator of a tendency.
-Point 5. Section 3. Materials and Methods.
Section 3.4.3. This part is a description of methods, please indicate the bibliographic sources
The bibliographic reference was added. Stashenko, E.E., , Jaramillo, B.E., , Martíınez, J.R. Comparison of different extraction methods for the analysis of volatile secondary metabolites of Lippia alba (Mill.) N.E. Brown, grown in Colombia, and evaluation of its in vitro antioxidant activity. 2004. Journal of Chromatography A, 1025: 93-103
Point 7. Please check the style for citations and references in all the paper
These citations were already checked.
Point 8. The number of bibliographic sources is adequate, but less than 35% of the total bibliographic sources are from the last 5 years.
You are right, but this can be an indicator that maybe some aspects of the screening methodologies should be reformulated, and this contribution could be an inciter and usuful tool for research in places with high biodiversity for bioprospecting.

Round 2
Reviewer 2 Report
The authors answered all the points I have asked in the first round of questions concerning this manuscript. However, one important aspect was not attended - the structure of alpha-pinene in reference 57 (not 54) is not the same presented in figure 2. Please, revise it and make adequate corrections.